# Facilitating interprofessional learning: experiences of using a digital activity for training handover of critically ill patients between a primary health care centre and ambulance services – a qualitative study

Marina Taloyan,[1,2] Conte Helen,[3] Åkesson Ninni,[4] Sofie Guldbrand,[5] Veronica Lindström  [6,7]

For numbered affiliations see end of article.

**Correspondence to**
Professor Veronica Lindström;
veronica.lindstrom@umu.se

## ABSTRACT

**Objective** To explore students' and facilitators' experiences of using a developed digital activity for interprofessional learning (IPL) focusing on critically ill patient handovers from a primary healthcare (PHC) centre to the ambulance service.

**Design** A qualitative study design was employed, and the reporting of this study adheres to the Consolidated criteria for Reporting Qualitative research guidelines for qualitative studies.

**Setting** A PHC centre and the ambulance service in Stockholm, Sweden.

**Participants** A total of 31 participants were included in the study: 22 students from five different healthcare professions, seven facilitators and two observers.

**Intervention** A digital IPL activity was developed to overcome geographical distances, and the scenario included the handover of a critically ill patient from personnel within the PHC centre to the ambulance service personnel for transport to an emergency department. Four digital IPL activities were conducted in 2021.

**Results** The digital IPL activity eliminated the issue of geographical distance for students and facilitators, and it enabled the students to find an interprofessional model for collaboration through reasoning, by communicating and sharing knowledge with the support of a common structure. Participants perceived the digital IPL activity and scenario as authentic, feasible and facilitated IPL. Using a case with an acute and life-threatening condition was a success factor for students to experience high realism in their IPL on patient safety, handover, care and treatment.

**Conclusion** The developed digital IPL activity facilitated the students' IPL and demonstrated potential sustainability as the digital approach supported overcoming geographical distances for both students and facilitators. By using a scenario involving an authentic case focusing on handovers of a critically ill patient, IPL, feasibility and acceptability were supported. However, it is crucial to emphasise that a comprehensive evaluation, both quantitative and qualitative, over an extended period of clinical rotations and involving a larger group of students

## STRENGTHS AND LIMITATIONS OF THIS STUDY

⇒ To our knowledge, this study is the first to collect data from digital interprofessional learning (IPL) learning activities with students in clinical rotations in both the primary healthcare centre and the ambulance service.
⇒ Multiple data sources (interviews, notes, observations) and 31 participants creating new knowledge based on qualitative analysis.
⇒ The number of participants and completed IPL activities could be considered low, and we do not know if all eligible students were reached.
⇒ Not all professions were representative among the students during all IPL activities.
⇒ The authors' preconceptions and personal experiences may have influenced the interpretation and understanding of the study.

is still warranted to ensure continuous improvement and development.

## INTRODUCTION

Establishing interprofessional learning (IPL) activities that ensure high-quality education is essential. This is particularly critical in clinical settings where IPL is still in the early stages of development. In the ambulance service, IPL is not yet firmly established. Historically, the ambulance service has been perceived as suboptimal for undergraduate students' clinical rotations, primarily due to the services' previous function of responding to emergency calls and conveying sick or injured individuals to hospitals. However, as the ambulance service has developed into a recognised discipline of pre-hospital emergency care, delivering advanced medical

interventions and care,[1] it has been shown that within the ambulance service, students can gain proficiency in nursing, medical and collaborative skills.[2] As the number of undergraduate nursing students completing clinical rotations in the ambulance service has increased, so have the expectations for activities to be conducted to support the students' IPL. However, different student professions undertaking clinical rotations in the ambulance service are limited. Therefore, in collaboration with other healthcare organisations such as primary healthcare (PHC) centre, there is a need to develop and systematically evaluate easily accessible IPL activities that are not dependent on the co-location of students. IPL activities have been developed and evaluated over the years, but the setting and context matter,[3] and contextual factors influence IPL.[3] Therefore, this study aimed to explore students' and facilitators' experiences using a developed digital activity for IPL focusing on critically ill patient handovers from the PHC centre to the ambulance service.

## BACKGROUND

IPL includes various activities depending on goals and desired outcomes, for example, short digital seminars or shadowing where the students' learning originates from interactive group processes, including collaborative analysis and joint critical reflection.[4] The students are active participants and learn with and from each other in authentic and patient-centred activities.[5] They negotiate their different experiences to fully understand their role and the roles of other professions.[6] However, finding places, spaces and time for students from various educational programmes to learn together in healthcare can be challenging since education is not always situated in contexts that support IPL.[7] Environments and care activities that naturally could support IPL are situated during clinical rotations where students from different educational programmes collaborate in patient care.[8 9] Students acknowledged that experiencing diverse IPL activities during clinical rotations is vital for learning how to work together.[10] Co-location is important but not enough for different professions to learn together.[10] Facilitated learning is part of building interprofessional rapport in groups, and balancing the participation of the different professions in their interactions is vital for learning together.[2 6 10] Students suggest that bringing a sense of their profession into the interactions and being inclusive of other professions are two key qualities for IPL[10] and that gaps in role conceptions are a crucial barrier[11] and could cause patient risks.

The WHO states that healthcare services should be safe and person-centred.[12] Research has shown that IPL activities can introduce patient safety concepts and experiential practice through interprofessional communication and analysis among students.[13 14] To improve patient safety in the healthcare setting, training must start with students.[15 16] Working as a professional in the ambulance service involves collaborating with various healthcare

professionals when providing care for patients transported to emergency departments or when responding to incidents in home healthcare settings.[17] There is limited knowledge about how undergraduate students perceive digital IPL activities focusing on handovers of critically ill patients within and between the PHC centre and the ambulance service. The handover between different healthcare settings is a complex process that includes communicating information between professionals responsible for the patient's care at different units or sites,[18] and communication failures are one of the leading causes of adverse events, particularly in handover situations.[19] When the handover is compromised, essential information can be altered or lost, transfer of cognitive bias may occur, patients may be exposed to adverse events, and patient safety issues may occur.[20] Expanding IPL activities beyond organisational boundaries to include the handover of patients between different healthcare organisations could allow students to practice and learn handover and patient safety in a new context. The ambulance service collaborates with PHC and community-based care and receives and hands over patients within these organisations.[21 22] To our knowledge, the handover of critically ill patients from the PHC centre to the ambulance service has not previously been used for IPL.

## METHODS

### Design

A qualitative study design was used to collect and analyse the students' and facilitators' experiences through their reflections on the newly developed digital IPL activity based on an authentic scenario. By examining the experiences of supervisors and students, conclusions can be reached regarding IPL in a sparsely explored IPL context, as well as the acceptability and feasibility of the developed digital IPL activity. Acceptability captures how the intended recipients react to the activity, that is, is it appropriate, relevant or sustainable,[23] and feasibility captures if the developed digital IPL activity is sustainable for continual use in the setting. The consolidated criteria for reporting qualitative research Consolidated criteria for Reporting Qualitative research guidelines guided the study.[24] The study was approved by the ethics review authority in Sweden (No.: 2021–00586).

### Patient and public involvement

No patient and public involvement.

### Setting

The PHC centre provides clinical rotations for medical, nursing, physiotherapy (PT), occupational therapy students (OT), and nurses studying at an advanced level to become primary healthcare nurses (PHCNs). In the PHC centre, the learning environment is characterised by care activities of acute and non-acute visits, short and long patient encounters, and the students care for patients with all varieties of complaints. The learning environment

varies between care for patients at home and the clinic; some patients are followed up on over time, and some only visit the clinic once. The PHC centre has established IPL activities but has never collaborated with the ambulance service concerning IPL activities for undergraduates. The ambulance service provides regular clinical rotations for nursing students and medical students more irregularly. The ambulance service responds to emergency calls, and the students care for patients with various complaints and acuity. Short patient care encounters characterise the learning environment in the ambulance service, care at patients' homes, at the scene of an injury and during transports between different healthcare facilities. The team in the ambulance consists of an emergency medical technician with basic life support competence and a registered nurse (RN) with 1 year of additional training in emergency care.[25] The clinical supervisor for both medical and nursing students in the ambulance service is the RN. Collaboration between PHC centres and the ambulance service primarily revolves around the handover of patients suffering from acute illness and needing to be transported from the PHC centre to an emergency department. With two different healthcare organisations collaborating and a lack of co-location, a prerequisite for conducting the IPL activity was that they were conducted digitally. In addition, using a digital format for the IPL activity increased the possibilities for students to participate and increased environmental sustainability by reducing unnecessary travel.

### Developing the Interprofessional learning activity and the scenario used

The scenario was developed by facilitators and researchers who were clinicians working in the ambulance service, PHC centre or intensive care unit. To ensure that students from ambulance services and PHC centre could collaborate and contribute with their various professional knowledge, the content of the scenario included (1) situations where different healthcare professionals cared for the patient, (2) handovers of patients in acute situations, and (3) patient safety in the context of PHC and the ambulance service. The scenario was built on several sources: (1) evidence-based guidelines[26–28], (2) perspectives of person-centred care[29], (3) the situation-background-assessment-recommendation (SBAR) communication tool[30] and finally, (4) crew resource management (CRM).[31] After developing the scenario, short sequences were recorded with a smartphone to illustrate different aspects of the care and handover situations. To reduce costs, the patient was acted by one of the facilitators and the recording was made by a clinical colleague from the ambulance service. In total, approximately six working hours were required for the development, filming and editing of the scenario and for embedding the material into a PowerPoint presentation. The scenario included critical events in the patient's symptom development when suffering from an acute myocardial infarction leading to a cardiac arrest and, after advanced cardio-pulmonary resuscitation (CPR), return to spontaneous circulation. The scenario also included handover situations and supplementary information (eg, vital signs and ECG). There was a balance between information and questions related to each healthcare profession to enable the students to learn from each profession during the scenario. Questions about CRM and SBAR were added in connection to each care transition in the scenario. The scenario included three episodes of handovers: (1) physiotherapist—primary healthcare nurse, (2) primary healthcare nurse or physician or physiotherapist—dispatcher at the emergency medical communication centre, and (3) physiotherapist or primary care nurse or physician—ambulance personnel. The IPL activity was completed within 150 min, including a 15-min break.

### Data collection and participants

The study and data collection were done in Stockholm, Sweden, where students did their clinical rotations in the ambulance service or at a PHC centre. The students invited to participate and evaluate the developed IPL activity were from three higher education institutions. The digital IPL activities were scheduled when there were eligible students from different educational programmes at their clinical rotations in the PHC centre or the ambulance service at the same time. The students were invited to participate in the IPL activity through e-mail and/or electronic bulletin boards and had no previous relationship with the researchers (VL, MT, HC). Some facilitators had prior knowledge about the students at the PHC centre as they also worked as clinical teachers (NÅ, SG), but they were not responsible for the students' grading. Data collection took place on four different occasions, once during the spring of 2021 and three times during the fall of 2021.

A combination of data sources was used to capture the participant's experiences in and on the activity. The students' and facilitators' reflective interplay during the IPL activity and group discussions after the activity were collected (a total of 12 hours of recording) using the digital platform recording function (image and sound). After each IPL activity, separate group discussions were held with the facilitation team. Data collection from the student and facilitator group discussions was initiated with an open-ended question ('What was your experience of this IPL activity?' vs 'What did you think about this activity?'). The facilitator's reflective notes were collected after each IPL activity.

In total, 22 students were consecutively included in the study, and seven facilitators and two observers participated in the IPL activity. The scenario was first piloted in the spring of 2021 with eight students, three interprofessional facilitators and two observers. The observers in this first scenario took notes on body language, group dynamics and reflections. The results of the analysis of the pilot led to minor adjustments with the addition of a normal ECG and reminders in the PowerPoint to discuss CRM and SBAR. The pilot was included in the analysis.

After piloting, three additional activities with 14 students from five educational programmes (medical, nursing, PT, OT and PHCN) were performed during the fall of 2021. In each activity, three facilitators participated. The interprofessional team of facilitators consisted of clinical teachers with previous experiences in IPL activities: two physical therapists, one specialist nurse with extensive expertise in the ambulance service, one specialising in paediatrics, one in primary healthcare and one physician. Participating facilitators in the activities were determined by the opportunity to participate. The recordings were transcribed verbatim, and the reflective notes were electronically transcribed.

### Analysis

The qualitative materials were analysed thematically, and the analysis was conducted according to the steps described by Kiger and Varpio.[32] Initially, the transcribed text and field notes from the four IPL activities were read multiple times to gain familiarity with the collected material. After this, codes were identified from the texts, and this step was followed by a search and construction of preliminary themes by sorting the identified codes. The preliminary themes were then refined and labelled. In the final step, the analysis was concluded by writing the results. No distinction was made in the analysis regarding either where the students did their clinical rotations or from which programme. Both medical and nursing students can do their clinical rotations in the ambulance service or the PHC centre. Neither were the quotes from the participants, distinguishing as this study aimed to explore on group levels both students' and facilitators' experiences of using a developed digital activity for IPL, focusing on critically ill patient handovers from a PHC centre to the ambulance service. The analysis process was iterative; a continuous movement between the codes, preliminary themes, and themes was made to preserve the essence of the collected data. Each step in the process was initiated by the last author and then reviewed by the co-authors. Any differences in interpretation were resolved through discussion until a consensus was reached.

## RESULTS

Three themes describing participants' experiences of IPL, the acceptability and the feasibility of the developed and used IPL activity, from both the students' and facilitators' perspectives, were constructed through the analysis: (1) interprofessional learning through collaborative reasoning in an authentic scenario, (2) sharing a common model by communicating information in a structured way and (3) reaching an understanding through collaborative reasoning.

### Interprofessional learning through collaborative reasoning in an authentic scenario

All participants discussed and evaluated the scenario as authentic and based on the patients' needs during handover in an acute situation. The students perceived the activity as a feasible and beneficial way of learning through analysing the patient's changing problems and needs between and with different professions throughout the whole chain of care. The students experienced that the IPL activity expanded their understanding since they usually did not get a chance to find out what happened with the patient before or after a care encounter. In addition, the students discussed that the developed scenario supported them in their discussions about the patient's follow-up care at PHC after an acute illness and the support needed for the patient's family.

> …a strength of today's seminar was that many different aspects of care, including discussions on the follow-up of the patient, became clear…(nurse student, activity #3)

The students agreed that all healthcare professionals need to know how to handle acute situations, for example, the CPR scenario. To enhance IPL during clinical rotations, students and facilitators also discussed the feasibility and possibilities of developing scenarios for future digital IPL activities by incorporating new film sequences into the scenario, for example, by changing symptom presentation and/or changing the outcome for the patient and thereby having the possibility to include other students in the IPL activity.

> Think about it, adding another outcome for the patient in the scenario can make it relevant for other students, eg, dietician or maybe homecare personnel (nurse facilitator 1, activity #3).

The facilitators, students and observers emphasised that sometimes, the medical focus took over the discussions between the students and risked reducing the possibilities for IPL. Facilitators explained several reasons for this. The first reason could be that not all healthcare professionals and care aspects were explicitly presented in the scenario and, therefore, were not discussed or analysed. The second reason was associated with the facilitators themselves.

> ….XXXX [name of facilitator] needs to take a step back, keep quiet and wait for the students to talk, discuss and analyse all different aspects of the care… (nurse facilitator 3, activity #2).

An additional reason contributing to the medical focus was the composition of the group of participants. When the group consisted of more medical students than nursing students, the discussions had more of a medical perspective. In contrast, when there were more OT or PT students in the group, their care perspective was discussed to a greater extent, and the medical issues in the scenario needed to be clarified more. In these situations, the facilitators needed to ensure that all aspects were discussed to ensure IPL and the acceptability of the developed scenario. During the scenario, the facilitators created a space for the students to take control, verbalise

and analyse together by listening, confirming and adding critical questions when needed. The facilitators encouraged the students to explain and describe their professional aspects when caring for the patient. Facilitators experienced initial uncertainty among the students and guided them to communicate by open and de-dramatising questions such as: 'Do you have any experience with something similar?…What good reflection do you have, anyone else? Don't think about right or wrong answers; start reasoning, and together we will find the solution' (PT facilitator 2 & nurse facilitator 3, activity #2,3,4)

### Sharing a common model by communicating information in a structured way

The students discussed how the scenario highlighted the importance of interprofessional communication in an acute situation and that all participants shared patient information in a structured way using SBAR. They described that using the same structure contributed to understanding how to communicate and reach a shared and patient-centred mental model.

> I know what to say, and the other person knows what to expect (PT student, activity #2).

The shared information is depending, on the other hand, on who they were communicating with and that participants were willing to listen to each other's point of view:

> You need to be an active listener…it doesn't matter what I say if they don't listen (PHCN student, activity #4).

Facilitators, observers and students commonly experienced using CRM to discuss leadership and professional responsibilities in care and in handover situations supported the students' learning of who is leading and who is responsible for which part of the care action. A conclusion was that situated leadership and responsibility should be taken by the person with the best knowledge of handling an acute situation regardless of professionally situated leadership.

> The one with the most experience and knowledge needs to take the lead in a cardiac arrest situation… I may not have the best knowledge… (medical student, activity #2) and a PT student continued in the same opinion: How to use everybody in a team when caring for a patient with a time-critical illness is essential (PT student, activity #1).

### Reaching an understanding through collaborative reasoning

According to students, the developed IPL activity and scenario facilitated both wider understanding and new knowledge through interprofessional collaborative reasoning. They learnt from each other about both the chain of care and the roles and responsibilities of other professions while switching between thinking and speaking from their perspective and listening to others in

planning care and medical treatment in both the ambulance service and at the PHC centre. One medical student expressed it as follows:

> The patient had been somewhere else before entering the PHC…, It [the scenario] had a holistic perspective …even though I know the physiotherapist profession, I did not understand their roles and responsibilities in the chain of care (medical student activity #2).

The students experienced the benefits of solving problems in an interprofessional team instead of doing it by themselves or with peers, as nurse students (activity #2) describe:

> …reasoning with different professions gave me more knowledge about what everyone can do and did in the care of the patient (nurse student, activity #2).

As students described, IPL in digital form added a new dimension and facilitated learning about each other's professions, knowledge, roles and responsibilities in different contexts without needing to go between different healthcare facilities. By verbalising their perspectives, listening to others, sharing knowledge and experiences, the students reached interprofessional conclusions regarding patient care. It can be illustrated by the following quotation:

> To do it [Scenario] interprofessional adds another dimension… that is very good… instead of just solving the case with my study colleagues … "… everyone has shared their knowledge… I have listened…and reflected on how to act in a similar situation in my context (OT student, activity #3).

The nurse students discussed that both CRM and SBAR also supported interprofessional reasoning when discussing the care for the patient during the different transitions in and between PHC centre and ambulance service:

> I have little experience of the emergency care… the discussions have been good to prepare myself… knowledge about whom I should turn to, who has the knowledge, what information I should pass on, what will they want to know (nurse student, activity #4).

The facilitators addressed the gap in the discussion when not all healthcare professionals were represented in the group of students who participated in the current study. In such cases, the facilitators took an active part in the discussion using their own clinical experiences to secure the authenticity and acceptability of the newly developed IPL activity. They contributed with their perspectives and knowledge to facilitate IPL and collaborative learning, including their learning from both the students and the other participating facilitators. However, the facilitators' participation was an act of balancing as the activity was developed for the students, and they needed the space to

discuss and solve the case in collaboration with the other students.

## DISCUSSION

The study explored IPL through the lens of collaborative reasoning using an authentic scenario presented in a digital IPL activity for students doing their clinical rotations in the ambulance service or at the PHC centre. A rationale for developing a digital IPL activity was the need for IPL activities in the ambulance service. By examining students, facilitators and observers' experiences of the activity, the study has shed light on the acceptability, feasibility and practical implications of incorporating a digital IPL activity for undergraduate students from different healthcare facilities, particularly in the context of critically ill patient handovers between PHC centre and the ambulance services. The findings show that digital IPL activity also positively impacts students' knowledge of and understanding of how to communicate by sharing information in a structured way, how to handle an acute care situation, and the value of collaborative reasoning. The scenario used in the IPL activity broadened all the students' perspectives on the chain of care and enhanced discussions and knowledge of various professions' roles and responsibilities. The use of a digital format of an IPL activity enabled learning across multiple geographical areas, which was necessary for the ambulance service, and in addition, underscored the importance of interprofessional problem-solving in handovers of patients within an acute care situation between different healthcare organisations. The findings also showed that when the activity was considered authentic, the students were able to communicate and share information, which enabled an interprofessional analysis. Therefore, it is reasonable to assume that the developed IPL activity supported students' learning through an interactive group process consisting of collaborative analysis and joint critical reflections, factors that are also described by Reeves *et al*[4] to support IPL. However, there is a need to balance the different professions in their interactions and for the students to maintain communication while facilitating learning and supporting interprofessional discussions.[2 6 10] With all the different healthcare professionals represented in the current IPL activity, the scenario covered different perspectives to achieve an interprofessional perspective on patient care through the entire care process. However, the different group compositions that could not be predicted before each IPL activity clarified the need for a scenario that could be used regardless of which professions participated in the IPL activity. Overall, it seemed that the digital IPL activity with a scenario including a patient with an acute illness was feasible and relevant for all healthcare professionals as they should be able to initiate care in an acute situation. However, out of this study, we do not know the effects of the uneven distribution among different health professions in the IPL activity; facilitators became more active and participated in the discussions by using their clinical profession when necessary. According to both facilitators and students, the facilitators directly impacted how the IPL activity was carried out. Students expressed a desire for more guidance and support from the facilitators, while facilitators recognised the importance of incorporating various professions' knowledge and perspectives into the clinical reasoning process for discussing optimal patient care. During the IPL activity, the facilitators employed different strategies to enhance the IPL. These strategies included encouraging active student participation, contributing additional knowledge to the scenario and discussions, asking questions to promote reflection and guiding students to summarise the discussions and patient findings using, for example, SBAR. The supervisors used facilitation strategies aligned with strategies identified in the Evans *et al* review.[33] The facilitators in the study did not represent all professionals of the students participating in all IPL activities, as suggested in previous literature.[34] In our study, when the facilitators did not represent all students' disciplines in the activity, they met before, after and during breaks and sent text messages during each activity to ensure that perspectives from all healthcare professionals were considered and discussed during the scenario. In addition, the meetings aimed to adjust the facilitators' different roles and strategies, as suggested in a previous study.[35] It may be considered important that the facilitator's healthcare professions represent all the various disciplines of students participating in an IPL activity, but it may also be of more importance that the students themselves represent the different disciplines in the discussions as described by Oosterom *et al*.[9] However, further research is needed to clarify whether facilitators must represent all the various disciplines of students participating in an IPL activity. Regardless, the ambulance service needs to collaborate with other healthcare organisations to be able to conduct IPL activities because the number of different professions within the organisation is limited. Finally, to our knowledge, this study is the first to describe a collaboration between a PHC centre and the ambulance service in developing a digital IPL activity for students. This collaboration can facilitate cooperation between two healthcare organisations to enhance safe patient care.

## CONCLUSION

The main findings of this study showed that the digital IPL activity and scenario developed were feasible for enhancing IPL related to acute situations, handover and patient safety in the context of PHC and the ambulance service. The positive impact of IPL concerning understanding and knowledge of different professions' perspectives on care and treatment in acute situations, particularly in the ambulance service and PHC context, was evident. The findings also underscored the central role of facilitators in promoting collaborative reasoning, emphasising the need for a balanced and inclusive approach for

optimal IPL outcomes. The developed digital IPL activity demonstrated potential sustainability and a necessity for the ambulance service as the digital approach supported overcoming geographical distances for both students and facilitators. By using a scenario involving an authentic case focusing on handovers of a critically ill patient, the feasibility and acceptability of using the developed digital activity for IPL were supported. By using digital solutions for the IPL activity, the ambulance service can offer IPL activities in collaboration with other healthcare settings for students doing their clinical rotations in the ambulance service. Finally, to our knowledge, this study is the first to describe a collaboration between a PHC centre and the ambulance service in developing a digital IPL activity for students. This collaboration can facilitate cooperation between two healthcare organisations to enhance safe patient care.

## Limitations

The study has several limitations. First, the number of participants and completed IPL activities could be considered low. All students doing their clinical rotation in the PHC centre or the ambulance service when the activities were conducted were invited by e-mail and/or electronic bulletin board, but we do not know if all eligible students were reached. Second, not all professions among the students were representative during IPL activities. This was adjusted for by the facilitators taking an active professional role in the IPL activity, which may have caused a decrease in students' participation in the discussions. However, this was not obvious during the analysis, but we do not know how the facilitators affected the IPL when the students did not represent all healthcare professionals. The third limitation of this study may have been the authors' preconceptions and personal experiences, which may have influenced the interpretation and understanding of the study. The fact that the authors are female and have extensive clinical experience in the ambulance service or the PHC, and the fifth author is an intensive care nurse and IPL promotor may introduce biases during the study design, data collection and analysis. Despite the potential for bias, the authors' preconceptions can also be a strength as they contribute to interpreting and understanding the study findings. Despite these limitations, this is the first study using data from IPL learning activities with students in clinical rotations in both the PHC centre and the ambulance service. Another strength of this study is that it was reasonable to assume that the IPL activity gave the students a possibility for IPL from the perspective of patient safety and patient needs in an acute situation. The third strength of the current study are multiple data sources (interviews, notes, observations) and 31 participants creating new knowledge based on qualitative analysis. Concerning the transferability of the study's findings, as the study is based on IPL, CRM, SBAR, guidelines and patient safety, considered as generic elements within healthcare services, it is reasonable to assume that the findings from this study are transferable to different contexts where interprofessional education and learning on acute situations and patient handovers take place. Using a digital activity for IPL was considered to facilitate sustainability, which can enhance transferability and collaboration among different healthcare providers with students doing their clinical rotations in different geographical areas. However, quantitative and qualitative evaluation is warranted for IPL learning activities with authentic scenarios in acute transition situations over a longer period of clinical rotations and with a higher number of students.

**Author affiliations**
[1]Department of Neurobiology, Care Sciences and Society, Division of Family Medicine and Primary Care, Karolinska Institute, Stockholm, Sweden
[2]Academic Primary Health Care Center, Region Stockholm, Stockholm, Sweden
[3]Department of Neurobiology, Care Sciences and Society, Division of Nursing, Karolinska Institutet, Stockholm, Sweden
[4]Rehab North West, Region Stockholm, Stockholm, Sweden
[5]Child Health Care Centre, Region Stockholm, Stockholm, Sweden
[6]Department of Nursing and the Ambulance Service, Västerbotten, Umeå University, Umea, Sweden
[7]Department of Health Promotion Science, Sophiahemmet Hogskola, Stockholm, Sweden

**Acknowledgements** Thank you to all the students who participated in this study, Kristina Björk sharing knowledge on care in the ambulance service, and academic clinical adjuncts Helena Solman, Hege Jahr, and Abdullah Almaasarani.

**Contributors** The first, second and last authors contributed during the study planning. HC and MT have a co-first authorship. VL, AN and SG collected data. VL initiated data analysis. HC and MT supported the data analysis. MT organised and coordinated learning activities at the Academic Primary Healthcare Centre. VL drafted the manuscript and all authors contributed, read and approved the final manuscript. VL is the guarantor and responsible for the overall content of this study.

**Funding** Karolinska Institutet, Stockholm, Sweden (educational funds for employees no.: 20200792), and Region Stockholm (ALF project no.: 20200223). No person from the founding committee participated in the study's design, data collection, analysis or writing of the manuscript.

**Competing interests** None declared.

**Patient and public involvement** Patients and/or the public were not involved in the design, or conduct, or reporting, or dissemination plans of this research.

**Patient consent for publication** Not applicable.

**Ethics approval** This study involves human participants and was approved by Etikprövnings myndigheten (The ethics review authority), Sweden (No.: 2021-00586). Participants gave informed consent to participate in the study before taking part.

**Provenance and peer review** Not commissioned; externally peer reviewed.

**Data availability statement** Data are available upon reasonable request. The data analysed during the current study are not publicly available due to the participants' privacy could be compromised but are available in Swedish from the corresponding author upon reasonable request.

**ORCID iD**
Veronica Lindström http://orcid.org/0000-0003-1386-3203

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
