## [Reviewer comments · BMJ Open]

ARTICLE DETAILS

TITLE (PROVISIONAL)	Facilitating Interprofessional learning: Experiences of using a digital activity for training handover of critically ill patients between a Primary Health Care Centre and Ambulance Services – a qualitative study
AUTHORS	Helen, Conte; Taloyan, Marina; Ninni, Åkesson; Guldbrand, Sofie; Lindström, Veronica

VERSION 1 – REVIEW

REVIEWER	Sakai, Ikuko Chiba University Graduate School of Nursing School of Nursing, Interprofessional Education Research Center
REVIEW RETURNED	15-Jan-2024

GENERAL COMMENTS	This is valuable and much-needed research. For further clarification, Please consider the following points. As well as examining the experiences of both students and facilitators, there should be a discussion of what the study aims to achieve. The summary is currently balanced but should be amended accordingly if the aims or considerations change. There appears to be a lack of a detailed description of this research programme. This makes it difficult to understand what experiences were gained from the learning objectives and learning content. In addition, although there is a description of the scenario setting, there is a lack of description of the environment in which this learning programme took place and what the setting was like. The discussion needs to briefly summarise the findings of this study, followed by a discussion in line with what the exploration of these experiences ultimately aims to achieve. In particular, an explanation of how and why learning programmes can be improved from these experiences is needed. The transferability of the research findings in comparison with previous studies also needs to be discussed.
---

REVIEWER	Han, Siew Ping Nanyang Technological University
REVIEW RETURNED	19-Jan-2024

GENERAL COMMENTS	The article is well structured with a clearly stated research question. The methods, results and discussion are aligned with the research question. However, the research question itself lacks novelty and focus. The authors aim to explore students' and facilitators' experiences of using a digital IPL activity focusing on acute patient handovers from PHC to EMS. In particular, they
---

	emphasised the novelty of studying IPL of the handover from PHC to EMS. However, they have not made a strong case for how IPL in this setting differs from IPL in general EMS settings, which has already been well-studied. The themes also do not provide new insight into aspects of the PHC-EMC transition or EMC IPL in general. The lack of focus can be seen in how the main point of each paragraph changes throughout the article. The introduction talks about the need for increased placement opportunities for students in EMS. The background talks about the importance of PHC-EMS transition. The results and discussion describe the dynamics of collaboration and communication. The conclusion focuses on the feasibility and authenticity of digital IPL. Overall, the analysis is mostly descriptive and generic without adding much to what has already been discussed in the field. The authors need to be clearer about what is the key message and consistently link all parts of the article to that central story. The research question and article as a whole could have been more focused if there was a theoretical framework to bring coherence to the story and provide more rigorous justification of how the current findings fill a gap in the existing literature. In my opinion, the use of digital approaches to overcome geographical and staffing constraints and the effect this has on authenticity could be a more novel and interesting take on the study, although of course this is by no means the only approach possible. What matters is that there is clear definition of the gap in knowledge and sufficient justification of how the current findings help to fill that gap. The standard of English is acceptable but lacks polish in places, particularly the Methods section. For example, there are sentences that are awkwardly phrased and/or contain slang: "Additionally, the acceptability of the digital IPL activity designed is an appropriate, relevant, or sustainable learning activity can be explored and described", "The students were invited to the IPL activity thru by e-mail and/or electronic bulletin board", "In each activity, three facilitators participated". There is potential for an interesting story here, but there needs to be more focus, coherence and theoretical backing.
--	---

VERSION 1 – AUTHOR RESPONSE

Reviewer #1 Dr. Ikuko Sakai, Chiba University Graduate School of Nursing School of Nursing	Dear Dr Ikuko Sakai thank you for the valuable comments supporting us in developing and clarifying the presentation of our study
As well as examining the experiences of both students and facilitators, there should be a discussion of what the study aims to achieve.	Thank you for the comment, information on what the study aims to achieve is added in the introduction and further discussed in the section on discussion

The summary is currently balanced but should be amended accordingly if the aims or considerations change.	A minor adjustment to the study aim is made and therefore some adjustments have been made in the conclusion and discussion
There appears to be a lack of a detailed description of this research program. This makes it difficult to understand what experiences were gained from the learning objectives and learning content.	Further information is added in the section on method.
In addition, although there is a description of the scenario setting, there is a lack of description of the environment in which this learning programme took place and what the setting was like.	Thank you for the remark, additional information about the setting and scenario is added in the section on method.
The discussion needs to briefly summarise the findings of this study, followed by a discussion in line with what the exploration of these experiences ultimately aims to achieve. In particular, an explanation of how and why learning programmes can be improved from these experiences is needed.	The discussion is developed in line with your suggestion.
The transferability of the research findings in comparison with previous studies also needs to be discussed.	Transferability is discussed in the section of limitations
Reviewer # 2 Dr. Siew Ping Han, Nanyang Technological University	Dear Dr Siew Ping Han, thank you for the valuable comments supporting us in developing and clarifying the presentation of our study
The research question itself lacks novelty and focus. The authors aim to explore students' and facilitators' experiences of using a digital IPL activity focusing on acute patient handovers from PHC to EMS. In particular, they emphasised the novelty of studying IPL of the handover from PHC to EMS. However, they have not made a strong case for how IPL in this setting differs from IPL in general EMS settings, which has already been well-studied. The themes also do not provide new insight into aspects of the PHC-EMC transition or EMC IPL in general.	Thank you for the comment, information on what the study aims to achieve is added in the introduction and further discussed in the result and limitation discussion
Overall, the analysis is mostly descriptive and generic without adding much to what has already been discussed in the field. The authors need to be clearer about what is the key message and consistently link all parts of the article to that central story.	Parts of the introduction, background, and discussion are re-formulated aiming to clarify our key message.

The research question and article as a whole could have been more focused if there was a theoretical framework to bring coherence to the story and provide more rigorous justification of how the current findings fill a gap in the existing literature. In my opinion, the use of digital approaches to overcome geographical and staffing constraints and the effect this has on authenticity could be a more novel and interesting take on the study, although of course this is by no means the only approach possible. What matters is that there is clear definition of the gap in knowledge and sufficient justification of how the current findings help to fill that gap.	Thank you for the comment, we have elaborated the discussion on what we assessed as novel in our study compared to previous knowledge and that is in line with your comment on the use of digital approaches to overcome geographical and staffing constraints.
The standard of English is acceptable but lacks polish in places, particularly the Methods section. For example, there are sentences that are awkwardly phrased and/or contain slang: "Additionally, the acceptability of the digital IPL activity designed is an appropriate, relevant, or sustainable learning activity can be explored and described", "The students were invited to the IPL activity thru by e-mail and/or electronic bulletin board", "In each activity, three facilitators participated".	The language is corrected and hopefully more polished.

VERSION 2 – REVIEW

REVIEWER	Han, Siew Ping Nanyang Technological University
REVIEW RETURNED	26-Apr-2024

GENERAL COMMENTS	I appreciate that the authors have taken on the suggestion to make a stronger case for the novelty of the study and highlight the use of digital approaches to overcome geographical and staffing constraints. However, this has only been incorporated in the changes to the introduction and discussion, with minimal changes to the analysis of the results. There is therefore now a lack of alignment between the themes discussed in the results and both the research question and the thrust of the discussion. Specifically, the themes are generic and do not highlight special features of training for ambulance services, and do not demonstrate how learning overcame geographic constraints. Therefore the key message still requires clarification. You may wish to refer to Lingard, Lorelei, and Chris Watling. "It's a Story, Not a Study: Writing an Effective Research Paper" (Academic Medicine 91, no. 12 (December 2016): e12–e12. https://doi.org/10.1097/ACM.0000000000001389) for how to structure a research article with coherence and relevance. Applying this to the current manuscript would require some re-analysis and substantial rewriting of the results section.
--

	Other than the above, the language is improved and there is better justification for the novelty of the study, although it could still be more convincing.
--	--

VERSION 2 – AUTHOR RESPONSE

Reviewer: 2

Dr. Siew Ping Han, Nanyang Technological University

Comments to the Author

Reviewer: I appreciate that the authors have taken on the suggestion to make a stronger case for the novelty of the study and highlight the use of digital approaches to overcome geographical and staffing constraints. However, this has only been incorporated in the changes to the introduction and discussion, with minimal changes to the analysis of the results. There is therefore now a lack of alignment between the themes discussed in the results and both the research question and the thrust of the discussion. Specifically, the themes are generic and do not highlight special features of training for ambulance services, and do not demonstrate how learning overcame geographic constraints.

Response:

We are thankful for your valuable comments and for pointing out important issues.

We have now clarified why the themes are considered generic: The study context used was considered suitable for IPL in the ambulance service as the scenario included handovers within and between one care unit to another. All students and facilitators participate throughout the entire IPL activity to learn from and with each other knowledge and experience in the handovers of the acutely ill patient. Therefore, the qualitative analysis includes the entire recorded and transcribed discussions among participants as one whole entity. The analysis is performed on a group level without classification of primary care or ambulance services; this is now clarified in the section on analysis. We have also added some sentences to highlight special features of training for ambulance services to clarify the alignment you suggested needed to be improved.

Furthermore, to strengthen the alignment throughout the whole manuscript we have added amendments in the manuscript regarding digital format as the following sentence on page 16: “The use of a digital format of an IPL activity enabled learning across multiple geographical areas, which was necessary for the ambulance service, and in addition, underscored the importance of interprofessional problem-solving in handovers of patients within an acute care situation between different healthcare organizations”.

Reviewer: Therefore the key message still requires clarification. You may wish to refer to Lingard, Lorelei, and Chris Watling. “It’s a Story, Not a Study: Writing an Effective Research Paper” (Academic Medicine 91, no. 12 (December 2016): e12–e12. <https://doi.org/10.1097/ACM.0000000000001389>) for how to structure a research article with coherence and relevance. Applying this to the current manuscript would require some re-analysis and substantial rewriting of the results section.

Response: Thank you for the suggestion

Reviewer: Other than the above, the language is improved and there is better justification for the novelty of the study, although it could still be more convincing.

Response: Thank you! We hope that you agree with all amendments to the manuscript.